# The Optical and Spectroscopic Properties of Fuchsite, Spodumene, and Lepidolite from Northern Scandinavia (Kautokeino, Kaustinen, Kolmozero)

**DOI:** 10.3390/ma16144894

**Published:** 2023-07-08

**Authors:** Miłosz Huber, Daniel M. Kamiński, Urszula Maciołek

**Affiliations:** 1Department of Geology, Soil Science and Geoinformacy, Faculty of Earth Science and Spatial Management, Maria Curie-Skłodowska University, 2d/107 Kraśnickie Rd., 20-718 Lublin, Poland; 2Faculty of Chemistry, Maria Curie-Skłodowska University, 2/508 Maria Curie-Skłodowska Sq., 20-031 Lublin, Poland; daniel.kaminski@mail.umcs.pl; 3Analytical Laboratory, Institute of Chemical Sciences, Faculty of Chemistry, Maria Curie-Skłodowska University, 3/27 Maria Curie-Skłodowska Sq., 20-031 Lublin, Poland; urszula.maciolek@mail.umcs.pl

**Keywords:** LCT pegmatites, Kautokeino, Kaustinen, Kolmozero, N Scandinavia, spodumene, fuchsite, lepidolite, crystal structure, optical properties

## Abstract

Li-Ce-Ta (LCT) pegmatites containing lithium mineralization in the form of spodumene and lepidolite, as well as fuchsite, from the regions of northern Scandinavia (N Norway, N Finland, N Russia) were studied. Detailed analyses of the chemical compositions of these minerals were carried out, involving scanning electron microscopy (SEM) with energy-dispersive spectroscopy (EDS), Fourier transform infrared (FTIR) spectroscopy with attenuated total reflectance (ATR), and X-ray photoelectron spectroscopy (XPS) studies. Their crystal structures were confirmed with the X-ray diffraction technique. Studies involving microscopy were also carried out, indicating the optical features of these minerals. Based on the analyses carried out in the studied rocks, the characteristics of these minerals were determined, as well as the crystallization conditions. This research indicates that the N Scandinavian area is prospective and may lead to further discoveries of this type of pegmatite in the studied region.

## 1. Introduction

In northern Scandinavia, pegmatite veins containing rare lithium and schists with chromium minerals are found in the regions of northern Norway [1], Finland [2], and the Kola Peninsula in Russia [3], among rocks constituting the cratonic basement of the Northern Fennoscandia [4]. These rocks contain minerals such as mica (lepidolite, fuchsite, muscovite) and pyroxene (spodumene), which co-occur with quartz, plagioclase, orthoclase, and several ore minerals: oxides (e.g., columbite, tantalite, cassiterite, sillenite, clarkeite) and sulfides (pyrite, chalcopyrite). Their thickness reaches several meters, and they cut through their metamorphic host rocks [5]. The previous study of these rocks was mainly concerned with analyses for the needs of economic geology [6]. The purpose of this article is a detailed analysis of the mineralogy of these rocks and a discussion of the crystalline structure characteristics of spodumene, lepidolite, and fuchsite.

## 2. Study Area and Massifs Geology

The northern part of the Scandinavian Peninsula is part of the Fennoscandia (Figure 1). In this region, Archean and Proterozoic rocks are exposed, forming various types of gneisses that constitute the basement of the Baltic Shield [3,7,8,9,10,11,12,13,14]. In the Fennoscandia, numerous intrusions have been found, such as the Tonalite Trondheimite Granodiorites of Murmansk (3.75 Ga) [3] and Jergul (2.97 Ga) [15], Patchenvarek anorthosites (2.92 Ga) [16], and enderbites (2.83 Ga) [9], with layered Paleoproterozoic intrusions in Monchegorsk (2.50 Ga) [17]. These rocks metamorphosed in the amphibolite and granulite facies [18,19]. Alkaline metavolcanic rocks and granitoid intrusions occur in the vicinity of the discussed pegmatite sites [20,21,22]. This is the case in the Kaustinen and Kolmozero areas, where numerous pegmatite veins can be found in these rocks [23,24,25,26]. In the Kaustinen area (Finland) and the Kolmozero area, there are pegmatite veins with lithium–cesium–tantalum mineralization (LCT), which are up to several meters thick and 100 m or more in length. In the Kautokeino area, schists containing chromic mica are visible. The age of Kautokeino and Kaustinen rocks spans the range of 1.9–1.8 Ga [27,28,29]. These rocks are exposed on the surface or covered by Pleistocene sediments [30,31]. In these locations, high-relief landscapes have developed [32,33,34,35,36].

## 3. Materials and Methods

Rock samples were collected between 2018 and 2022. During this period, the discussed massifs were visited, and the geological documentation of the samples was also carried out. In Scandinavia, rock samples were taken from the site of their occurrence. The selected rocks were targeted for thin-section preparations. Subsequently, the minerals were subjected to analyses using a Leica DM2500P (Heerbrugg, Switzerland) polarizing optical microscope [38,39,40,41] and with a scanning electron microscope, a Hitachi SU6600 (Tokyo, Japan), equipped with energy-dispersive spectroscopy (EDS) [42,43]. The samples were analyzed under a low vacuum (10 Pa), 15 kV accelerating voltage, and beam diameter of 0.2 µm. A total of 525 analyses were performed using the microprobe (at Kautokeino, 72; at Kaustinen, 179; and at Kolmozero, 274). The selected minerals were separated and analyzed with single-crystal X-ray diffraction. Data were collected using a Rigaku diffractometer (Tokyo, Japan) [44] with CuKα radiation (λ = 1.54184 Å) at 293 K. Crystallographic refinement and data collection, as well as data reduction and analysis, were performed with a CrysAlisPro v42 [45]. The selected single crystals were mounted on the nylon loop with oil. The structures were determined by applying direct methods using the SHELXS-86 program and refined with SHELXL [46,47] in Olex2 software [48]. Table 1 provides the experimental details for the single crystals’ X-ray measurements. These samples were also examined with the Fourier transform infrared (FTIR) technique [49,50]. The samples were measured using a Nicolet 8700A Thermo Scientific Shimadzu FTIR spectrophotometer (Waltham, MA, USA) equipped with an attenuated total reflectance (ATR) accessory, a Smart Orbit™ diamond ATR QATR-S (Riyadh, Saudi Arabia) (wideband diamond crystal). Every spectrum was obtained from 256 records at a 4 cm^−1^ resolution ranging from 4000 to 400 cm^−1^. The elaboration of a spectrum was carried out using the OMNIC program. Optical and microscopic studies were performed at the Department of Geology, Soil Science, and Geoinformation of the Institute of Earth and Environmental Sciences, and crystal chemistry studies were performed at the Department of Crystallochemistry, Faculty of Chemistry, Maria Curie-Skłodowska University, in Lublin. The UHV multi-chamber system (Prevac, Poland) was used to carry out the X-ray photoelectron spectroscopy (XPS) measurements. Molybdenum mounts were used to hold the measured samples. The chamber pressure was 5 × 10^−9^ mbar during analysis. The mineral surface was excited with X-rays (Al Kα, 1486.6 eV) from a Scienta SAX-100 X-ray source equipped with an XM 650 X-ray monochromator. The hemispherical electron analyzer R 4000 (Scienta, Uppsala, Sweden) operating in sweep mode was used to detect photons arriving from the sample. The energy of the spectra was calibrated using the C1s aliphatic carbon peak, EB = 285 eV. The CasaXPS v2.3.23-PR1 software from Casa Software Ltd. (London, UK) was used to analyze the measured data. The full width at half maximum (FWHM) and relative peak shift were fixed in the fitting process [51,52].

## 4. Results

### 4.1. Host Rock Petrology

The examined lithologies are located in the Kolmozero–Voronya Greenstone Belt, Pohjanmaa Schist Belt (Kaustinen Pegmatites), and Kautokeino Greenstone Belt. The LCT pegmatites from the Kolmozero area are located in the Kolmozero–Voronya Greenstone Belt, separating the Murmansk block from the Kola block, composed of mixed metasediments and metavolcanites. These formations are intersected by numerous intrusions of alkaline and acidic rocks. The discussed LCT pegmatites intersect the alkaline rocks of the Patchemvarek anorthosite intrusion. This intrusion is located on the border of the Kolmozero block with the plagiogranites of the Murmansk block [53]. These rocks are in direct contact with biotite gneisses and amphibolites of the Kolmozero–Voronya Belt. There are fine-grained tourmaline-muscovite granites with pegmatite apophyses rich in tourmaline, garnet, and apatite. The studies of Kudrashov et al. [54] indicated that these are pegmatites of hydrothermal–metasomatic origin [55,56].

The Kaustinen pegmatite province is located in Western Finland. It is situated among supracrustal rocks belonging to the Pohjanmaa Schist Belt [57,58]. It is surrounded by the Vaasa granitoid complex to the west and Central Finland granitoides to the east [6]. The Pohjanmaa Schist Belt is composed of micaceous schists and gneisses, along with metavolcanic rocks. These rocks were metamorphosed under the amphibolite facies of 1.89–1.88 Ga [59]. The lithium-rich pegmatites from the Kaustinen province belong to the albite spodumene type according to the classification of Černy and Ercit [55]. They were formed during the metamorphism of the rocks of the Pohjanmaa belt. The pegmatites form a complex of veins cutting the rocks of the Pohjanmaa Käpyaho belt and others [60,61,62,63].

In the region of the Palaeoproterozoic Kautokeino Greenstone Belt, where muscovite-fuchsite and quartz-orthoclase schists are located between the gneissic Ráiseatnu Complex (1868–1828 Ma) to the west and the metaplutonic Jergul Complex to the east (tonalite–trondhjemite–granodiorite–granite plutonic rocks formed between 2975 and 2776 Ma). In the Kautokeino Greenstone Belt, metasedimentary–metavolcanic rocks are present, with numerous mafic intrusions [27]. The Masi Formation in the Archean basement is formed of a quartz-feldspar conglomerate with muscovite interbedding and ore mineralization composed of iron and copper sulfides (pyrite, chalcopyrite). It is intersected by the mafic sills of the Haaskalehto formation (2220 Ma age). [64] Among these formations are the discussed fuchsite-rich rocks, which may have formed as an alteration product of detrital chromite grains [65].

#### 4.1.1. Kautokeino

The paleoproterozoic metamorphic formations classified as the Alta-Kautokeino Greenstone Belt are exposed in the Kautokeino region. Among these formations, there are mica schists [66] containing fuchsite (Figure 2A). This rock is an intense green-pink, characterized by layering resulting from a gneissic, streaky texture. The rock has a grano-lepidoblastic texture and is also characterized by a glomeroblastic, locally diablastic texture. Under an optical microscope, large quartz crystals are visible, forming irregular, hooked aggregates in contact with each other. Opaque minerals, such as pyrite and chalcopyrite, are visible between the muscovite and fuchsite. Close to the quartz crystals, microcline and Na-rich plagioclase are also present, forming leucocratic zones. In the interstices of plagioclase and microcline, small crystals of epidote are encountered. In addition to these zones are areas richer in femic minerals. They are represented by aggregates of biotite, which co-occur with fuchsite to form a streaky structure in the rock. These minerals form aggregates, resulting in interlaced streaks in the discussed rock. Opaque minerals can be observed, and zircon and apatite are woven into the biotite flakes. The detailed results of the phase studies in the micro-area are discussed below.

#### 4.1.2. Kaustinen

In the Kaustinen area, pegmatites with a silicon-gray color, coarse crystalline structure, and compact, disordered texture are exposed among the gneisses. Their age was determined to be 1.79 Ga (U-Pb method) [67]. Under an optical microscope, the quartz crystals form large grains in contact with each other and closely interlocking. The quartz in the studied rock has wavy extinction. Alongside these are large orthoclase crystals adjacent to the quartz (Figure 2B). They are accompanied by much smaller tabular grains of Na-rich plagioclase, often between the orthoclase and quartz. Femic minerals are represented by muscovite forming large clusters of flakes, in the background of which fine zircon crystals are visible. Muscovite crystals are near the pyroxene. Biotite is also visible in the form of small flakes, usually between large pyroxene crystals. The pyroxenes in the discussed rock form large crystals, reaching several centimeters in size. They are represented by spodumene (and hypersthene). The spodumene forms compact crystals with jagged boundaries in which quartz and muscovite are visible. The hypersthene is anhedral. Opaque minerals (columbite-tantalite, sillenite) are visible in the form of small crystals close to the quartz, sometimes also forming solid inclusions, accompanying zircon. The detailed results of the micro-area phase studies are discussed below.

#### 4.1.3. Kolmozero

In the area of the Kola Peninsula between the Kola and Murmansk blocks is the Archean Kolmozero–Voronya Greenstone Belt. Adjacent to the LCT pegmatites are Archean gabbro-anorthosites and granitic rocks. The analyzed pegmatites are dated to 1.90–1.86 Ga [68]. The pegmatites containing spodumene and lepidolite are cream-gray-colored rocks with a coarse crystalline texture and a compact, disorderly texture (Figure 2C). In thin sections, there are large crystals of quartz interlocking with each other. The quartz forms clumped aggregates in the space between the other phases. Alongside these minerals are visible crystals of Na-rich plagioclase. In the described rock, orthoclase, usually reaching a considerable size, is also visible near the other leucocratic minerals. Among the potassium feldspars, microcline usually forms small crystals. Alongside these minerals, flakes of biotite and muscovite can be identified, interspersed among the quartz and plagioclase. Biotite is much less abundant than muscovite, which predominates. In the pegmatites with lepidolite, this mineral accompanies muscovite, forming deformed flake aggregates of varying sizes. Large crystals of spodumene, reaching several centimeters in length, are visible alongside these minerals. They are surrounded by biotite, lepidolite, and plagioclase. In addition, small crystals of opaque minerals are visible near muscovite flakes occurring in interstices of quartz and feldspar. Zircon crystals are also visible within mica flakes. Accessory apatite forms small crystals co-occurring with femic minerals.

### 4.2. SEM-EDS Analyses

In the case of the Kautokeino rocks, the examined trioctahedral mica can mainly be classified as annite and siderophyllite [69,70,71] (Figure 3). These micas co-occur with fuchsite-forming overgrowths (Table A1).

Dioctahedral micas are represented by muscovite with fuchsite and lepidolite (Table A1). Muscovite usually shows a low Na^+^ content (up to 5 wt.%). In the rocks from Kautokeino, in addition to muscovite, it was found that all the chromic micas examined were fuchsites. Some of the examined chromium micas have a composition characteristic of paragonite (Figure 4). In addition, an admixture of clinozoisite and epidote (mainly in the Kaustinen pegmatites, Table A1) was found in the discussed rocks.

The examined pyroxenes are mainly represented by spodumene, with an Na content of up to 5 wt.%. In the examined samples, spodumene dominates, while admixtures of jade particles occur in small amounts (Table A2). Hypersthene was also found in the latter.

The accompanying leucocratic minerals are represented by quartz, usually with an admixture of up to 4% of aluminum oxide. The plagioclase is represented by albite with a small admixture of oligoclase (4%), as well as andesine (6%) and labradorite (6%). The labradorite is an admixture found mainly in the pegmatites from Kaustinen. Accessory minerals are represented by zircon and apatite. Zircon crystals are mainly found in close association with muscovite flakes. Phosphates are represented mainly by a variety of hydroxyapatites, with approximately 3% carboxy apatite and 2% fluoroapatite.

In addition, opaque minerals were found in the exanimated rocks. In the Kolmozero pegmatite, columbite ((Fe,Mn)Nb_2_O_6_) forms euhedral crystals, usually close to spodumene, quartz, and plagioclase [2]. Along with columbite, tantalite is visible. Tantalite ((Fe, Mn)Ta_2_O_6_) is replaced with bixbyite in the oxidation zone. Magnetite was also found. The sample also presents clarkeite (Na,Ca,Pb)_2_(UO2)_2_(O,OH)_3_ in the vicinity of apatite and sillenite Bi_12_SiO in the vicinity of femic minerals.

The opaque minerals in the investigated rocks are represented by multiple phases. In the case of the schists from Kautokeino, the opaque phases include cassiterite and pyrite. In the pegmatite from Kaustinen, the opaque phases include magnetite and ilmenite, accompanied by titanite. Trace or minor cassiterite was also found, as well as galena, sphalerite, and chalcopyrite, which, together with barite, form disseminations. Columbite and tantalite were also found, although in smaller amounts relative to the Kolmozero pegmatite.

### 4.3. Optical Properties of the Discussed Minerals

Spodumene forms large xenomorphic crystals, with sizes reaching several cm. This mineral usually has a light green color. Sometimes, it resembles plagioclase, which, when undergoing sericitization, also has a slight gray-green tint. In the microscopic image, it sometimes forms a diablastic texture according to the (100) miller index. It is usually found near micas represented by muscovite, lepidolite, and, less often, biotite (Figure 5). The schist is highly visible. Small admixtures of Fe-oxides can be seen along the schistosity of the rock. The straw color on the thin section has a clear, positive relief, with darkening extinction.

Lepidolite forms large lamellar aggregates, usually colored pearly pink. Macroscopically, this mineral forms flakes reaching up to 1 cm in size. It is usually quite visible in the rock due to its coloration and luster. In the microscopic image, it forms numerous adhesions of varying sizes. It is present with spodumene, near plagioclase and quartz. Between the mica flakes, rutile be observed. In the thin section, it is colorless, with a faint, negative relief. Under polarized light, it shows second-order interference colors, optically resembling biotite (Figure 6). Our microscopic observations of lepidolite showed some deformation of its lamellae due to dynamic processes.

Fuchsite forms fine, scaly accumulations, the size of which reaches several millimeters. Macroscopically, it is colored green and has a pearly luster. It is highly visible against the background of biotite and feldspar in the investigated rocks. Fine zircon grains are visible in the background of the fuchsite aggregates (Figure 7). In the microscopic image, they form polysynthetic adhesions with muscovite occurring between quartz and orthoclase. The sample shows pleochroism with a delicate greenish (β)-bluish (α) coloration. Under polarized light, it shows intense second-order interference colors, making it similar to muscovite.

### 4.4. Spectroscopic Properties of the Minerals under Investigation

Infrared studies carried out for spodumene showed some small oscillations at a wavelength of 3612 cm^−1^, which can be explained by the influence of water. Values in the vicinity of 1005 cm^−1^ may be related to stretching vibrations for silicon-oxygen tetrahedra [74]. Vibrations in the vicinity of 779 cm^−1^ can be correlated with Si-O stretching vibrations. Similarly, for lengths of 647 cm^−1^ to 448 cm^−1^, non-bridging bending vibrations for O-Si-O can be found with the participation of aluminum, which can also substitute the spodumene structure in an octahedral position (Figure 8). The latter oscillations are also affected by the position of lithium (448 cm^−1^), which, combined with oxygen, contributes to their modification. Through a comparison with the crystals of jadeite, it can be seen that substitution with Na cation with an ionic radius of 186 pm [75] in the M6 position shifts these vibrations to 455 cm^−1^. In comparison, magnesium enstatite, in the M6 position, has 447 cm^−1^ vibrations (enstatite [76] and bronzite [77]). Lithium is a much lighter element than sodium, as its molar mass is 6.941 (for sodium it is 22.989 [51,78] close parenthesis g/mol) and its ionic radius is 152 pm. The full width at half maximum of the 455 cm^−1^ vibrations suggests that these vibrations are partly derived from the sodium at this position.

In the case of the study micas (fuchsite and lepidolite, Figure 9), comparisons were made with muscovite [79]. In both cases, the absorbance characteristic of stretching vibrations of the OH groups in the vicinity of 3625 and 3608 cm^−^^1^ is visible (Figure 9). These are determined by vibrations between ions located in octahedral groups and their interaction with water [80,81]. These differences become apparent depending on the nature of the ions in the analyzed minerals (Li in lepidolite, Cr in fuchsite). There is also a slight increase in absorbance in the region of 1621 cm^−^^1^, which is more pronounced for lepidolite. Another oscillation in the region of 970 cm^−^^1^ and 960 cm^−^^1^ is related to deformations produced through connection between aluminum and the OH group in these minerals [82]. In the region of 796–799 cm^−^^1^, there are Si-O deformations in both mica, and those at 750 cm^−^^1^ are characteristic of O-Al-O stretching vibrations in the tetrahedral position. Similarly, in lepidolite, a band in the 523 cm^−^^1^ region was found that is characteristic of Al(Li)-O-Si vibrations. The last absorbance in the 464 cm^−^^1^ and 441 cm^−^^1^ is associated with vibrations in the Si-O-Si group [82].

### 4.5. Single Crystals and XPS Analysis Results

#### 4.5.1. Fuchsite

The studied fuchsite crystals, like all minerals in this group, contain chromium [82,83]. This is visible in the results of the diffraction measurements for the monocrystals and XPS. The X-ray data indicate that the studied fuchsite crystal contains 0.4 ions of this element per elemental cell of the crystal in the Cr-Al layer (Figure 7). It is worth noting the staggered position of both the Al^3+^ and Cr^3+^ ions at a distance of 0.12(1) Å (Figure 10 and Figure 11). This spread is due to the difference in the ionic radii of these two elements. The diffraction data show that only one position in the lattice, where the Al^3+^ ions are present, is occupied by additional Cr^3+^ ions, while the other position is 100% occupied by Al^3+^ ions (Figure 10). In addition, the layer has a free space filled, in this case, with water or an OH^-^ group in the amount of 0.1 molecule/ion per elementary cell. In this case, the OH^-^ group can compensate for the positive charge of the cations. In the Si-K layer, no admixture of other ions is observed at the detection level of this technique. All cell parameters and compositions are within the typical limits for this type of mineral. The thermal vibration ellipsoids observed in the experiment at 296 K had small thermal vibration amplitudes of U_iso_ 0.02–0.03 (Figure 11). This indicates strong interactions between ions in the lattice.

#### 4.5.2. Lepidolite

Diffraction studies of the lepidolite yielded its structure [84,85,86,87]. They confirmed that it is a mineral from the silicate cluster, classified as lithium micas with an admixture of Fe^2+^ or Mn^2+^ in one position. However, unequivocally determining which ion is in this position is impossible with this technique. Positively charged Li-K ions reside in the space between the negatively charged aluminosilicate layers. In addition, OH^−^ group or water molecules in the amount of 0.1 per elementary cell of the crystal can be found in this crystal net space (Figure 12). In an elementary cell, there are three such layers in the direction of *b*. A richer atomic composition is provided by the data obtained through XPS measurements. This technique is more sensitive to heavier elements and allows for the unambiguous determination of their type. In contrast, lithium ions, which are difficult to determine via XPS, are visible in the monocrystalline structure. The ratio of K/Li ions is 2:1 in the structure. A close fit of the X-ray data with the model indicates that the lattice occupancy of Al^3+^ ions is 90 ± 5%. This indicates that aluminum shares this position with lighter ions, e.g., Mg and Na. However, (Figure 10), in this case, it is necessary to properly balance the charges in the lattice. Also, the occupancy of the Li position is less than 100%, which, in this case, may indicate the partial substitution of this position with water molecules. The low values of the Q_izo_ parameters (0.02–0.03) for atoms at 296 K indicate strong interactions between atoms (Figure 13).

#### 4.5.3. Spodumene

This is another studied mineral from the silicate cluster containing lithium [88,89,90,91]. In this case, the structural studies of the crystals indicated that the stoichiometric contents of Li and Al in the crystal lattice are within the error margin. Attempts to fit a crystal lattice model with a free occupancy parameter for Li indicated that the modeled electron density at this site is slightly higher than 1, at 1.06 (Figure 14). This may indicate a small content of a heavier element at this site, e.g., sodium, as suggested by the XPS and FTIR measurements. In the studied monocrystals, other ions were not visible in the crystal lattice (Figure 12). In the spodumene, silicate ions connect adjacent layers, so there is not enough space for additional ions between them, which explains the composition of this mineral. The compact lattice structure also contributes to strong interactions between atoms, which, in turn, manifests itself in the lowest Q_izo_ values of all the minerals presented (0.006–0.017) and its highest relative hardness (Figure 15).

## 5. Discussion

The minerals included in rocks found in northern Scandinavia (within Norway, Finland, and Russia) [92,93,94,95,96,97] were studied. These are exposed in many places in the area under discussion. Due to their mineralization, they may be significant in terms of raw material development, although the relatively small size of these pegmatite veins makes their profitability highly dependent on raw material prices. The mineralization of these pegmatites is the result of the crystallization of residual melts, which contain many incompatible elements [55,98]. On the other hand, the presence of some elements is related to the chemistry of the host rocks. This is particularly evident in the case of fuchsite, in which the presence of chromium may be related to specific rock types. Such small occurrences of fuchsite have been found by the authors in vein rocks in the Monchepluton area (Russia). Lepidolite and spodumene, on the other hand, show small admixtures of sodium, manganese, and iron, which may also be related to the rocks near these pegmatites. This was confirmed by both micro-area studies and monocrystal and XPS analyses. The presence of water in the mica and even in the spodumene (found using FTIR) confirms the hydrothermal nature of the association between these minerals [99,100]. Their nature in the discussed rocks varies. Optical studies indicated that the spodumene in the analyzed pegmatites usually has hypidiomorphic crystals, which may indicate crystallization as an early silicate phase. The presence of small admixtures of sodium may indicate that the mineral originally formed at a great depth, where higher pressures prevail, and then tectonically dislocated with solutions to its present site of occurrence, where it was hydrothermally altered [53,54,57]. On the other hand, aggregates of fuchsite and lepidolite tend to be secondary, co-occurring with other micas and occupying a position in the interstices of existing minerals; moreover, lepidolite can probably be a secondary mineral, formed at the expense of spodumene, as evidenced by our observations of this mineral in pegmatites. It is noteworthy that in addition to these minerals, there are many accessory phases, such as apatite and zircon. Alongside these, opaque minerals are present in large numbers. The presence of tin minerals points to the granitoid association as the source material for the origin of the schists [101,102]. In the case of the LCT pegmatites found in the Kaustinen area, these include, in addition to the aforementioned cassiterite and barite, magnetite, ilmenite, and titanite, as well as galena, sphalerite, and chalcopyrite. The presence of these minerals may also be related to the granitoid products and the action of hydrothermal products. The mineralization of Kolmozero pegmatites is also associated with the action of hydrothermal formations. This is evidenced by the mutual relations of rock-forming minerals with the observed bixbite, clarkeite, and sillenite, which are also accompanied by magnetite. Columbite and tantalite were also found in the LCT pegmatites.

## 6. Conclusions

The studied minerals are from selected rocks that are exposed in northern Scandinavia. The studied mineralization indicates the hydrothermal nature of these components. The studied minerals, including spodumene, lepidolite, and fuchsite, were formed after residual crystallization. The small admixtures of sodium present in the spodumene may indicate that it was formed under high pressures and, together with the melt, reached its present location, where it was altered due to pressure from hydrothermal fluids. The occurring lepidolite crystallized at a later stage, at the expense of spodumene, as evidenced by the structure of this mineral in the rock and by association (the occurrence of relics of spodumene in the vicinity of lepidolite). Fuchsite, like lepidolite, crystallized in the final stage, co-occurring with muscovite and biotite. The presence of chromium ions in this fuchsite is probably due to the occurrence of chromium-containing rocks in the vicinity of the formation of intrusions from the residual melts in which the studied rocks crystallized. The presence of accessory and opaque minerals also attests to the granitoid association of the original solutions. The minerals present may be of economic importance. Thorough research may contribute to the discovery of new locations of this type of pegmatite in the discussed area.

## Figures and Tables

**Figure 1 materials-16-04894-f001:**
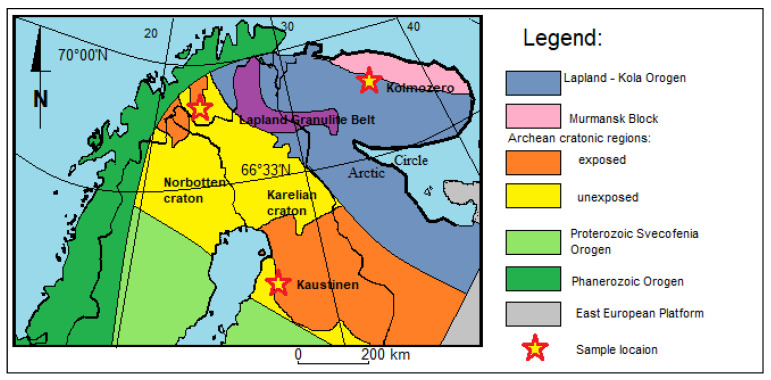
Simplified geological–tectonic map of N Scandinavia (after [37]) with the sample locations.

**Figure 2 materials-16-04894-f002:**
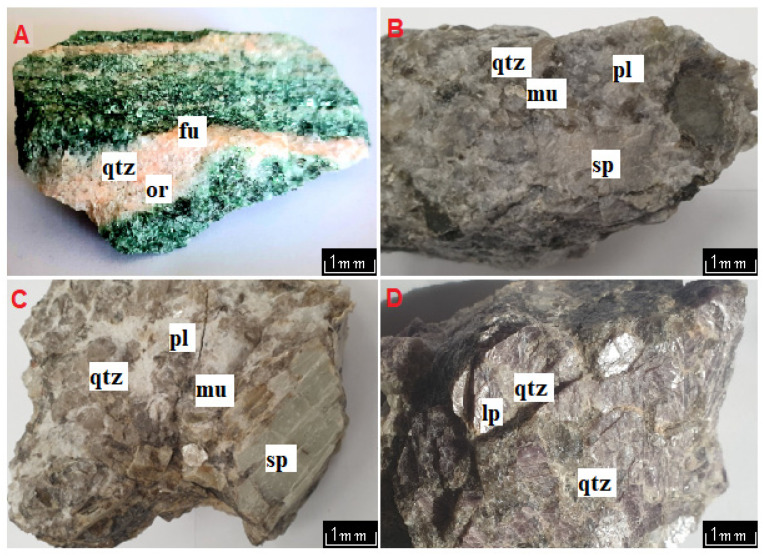
Macroscopic images of rocks under investigation. Schist with fuchsite from Kautokeino (**A**), pegmatite with lepidolite from Kaustinen (**B**), and pegmatite from Kolmozero with spodumene (**C**) and lepidolite (**D**). Abbreviations: fu—fuchsite, sp—spodumene, lp—lepidolite, qtz—quartz, mu—muscovie, pl—plagioclase, or—orthoclase.

**Figure 3 materials-16-04894-f003:**
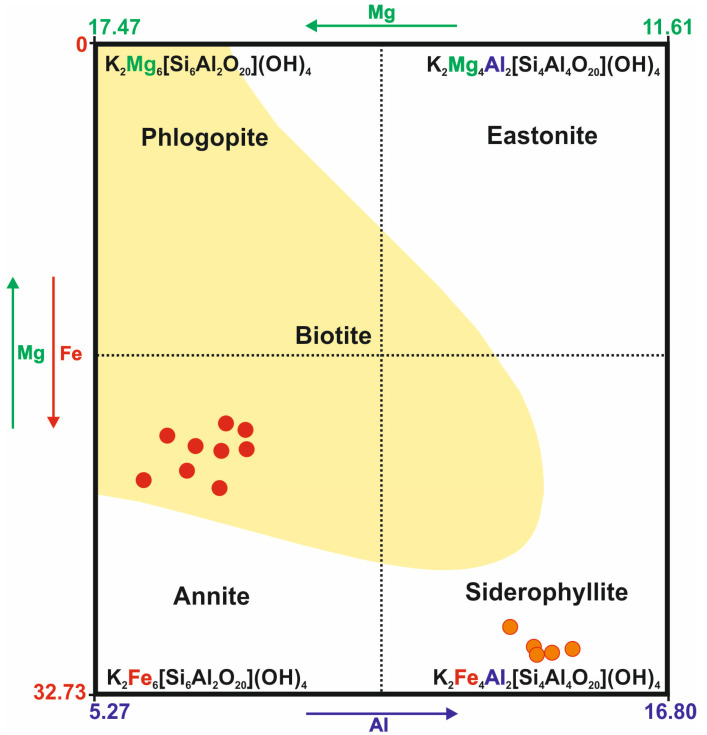
Mineral composition of trioctahedral micas from Kautokeino schists. The element content is expressed in wt.%. The yellow area represents the composition of the most frequent, naturally occurring biotites [71].

**Figure 4 materials-16-04894-f004:**
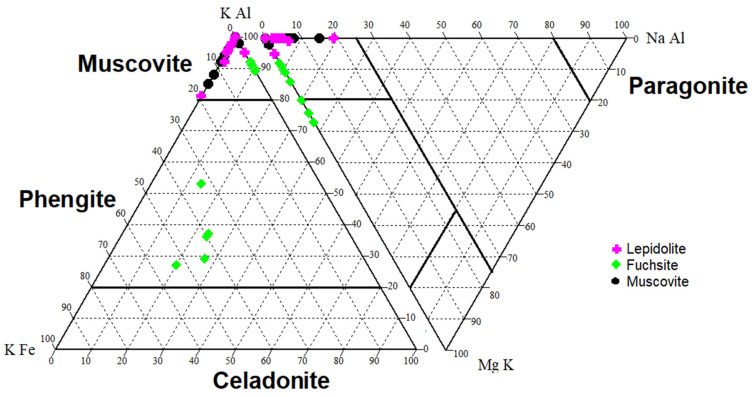
Type of dioctahedral micas examined in the discussed rocks (based on [72,73], modified by the authors). The element content is expressed in wt.%.

**Figure 5 materials-16-04894-f005:**
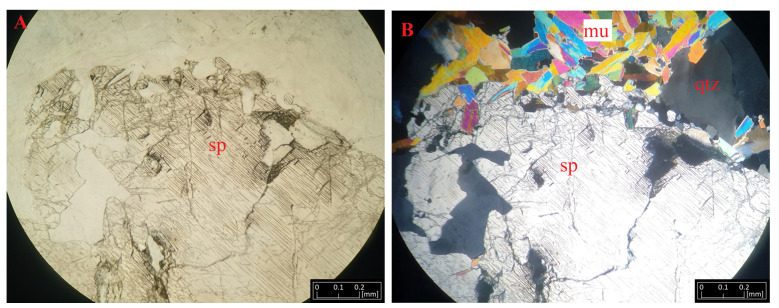
Microphotographs of spodumene (sp) from Kaustinen under transmitted light (**A**) and polarized light (**B**) accompanying quartz (qtz) and muscovite (mu) crystals.

**Figure 6 materials-16-04894-f006:**
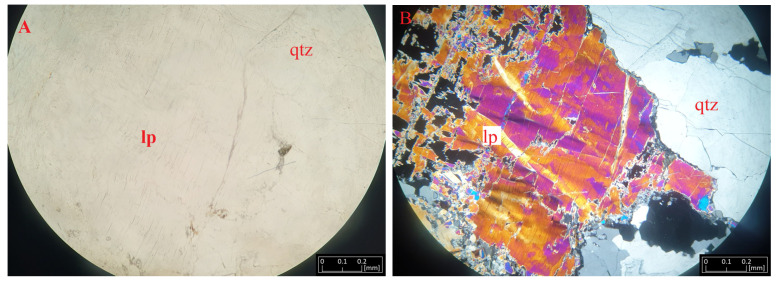
Microphotographs of lepidolite (lp) from Kolmozero under transmitted light (**A**) and polarized light (**B**) with quartz (qtz).

**Figure 7 materials-16-04894-f007:**
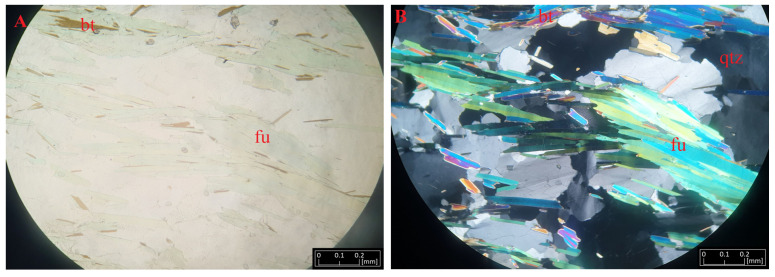
Microphotographs of fuchsite (fu) from Kautoteino under transmitted light (**A**) and polarized light (**B**) near biotite (bt) and quartz (qtz) crystals.

**Figure 8 materials-16-04894-f008:**
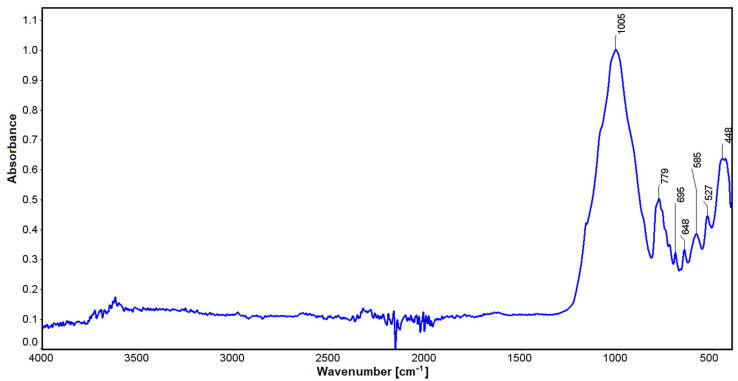
IR spectrum for the Kaustinen spodumene.

**Figure 9 materials-16-04894-f009:**
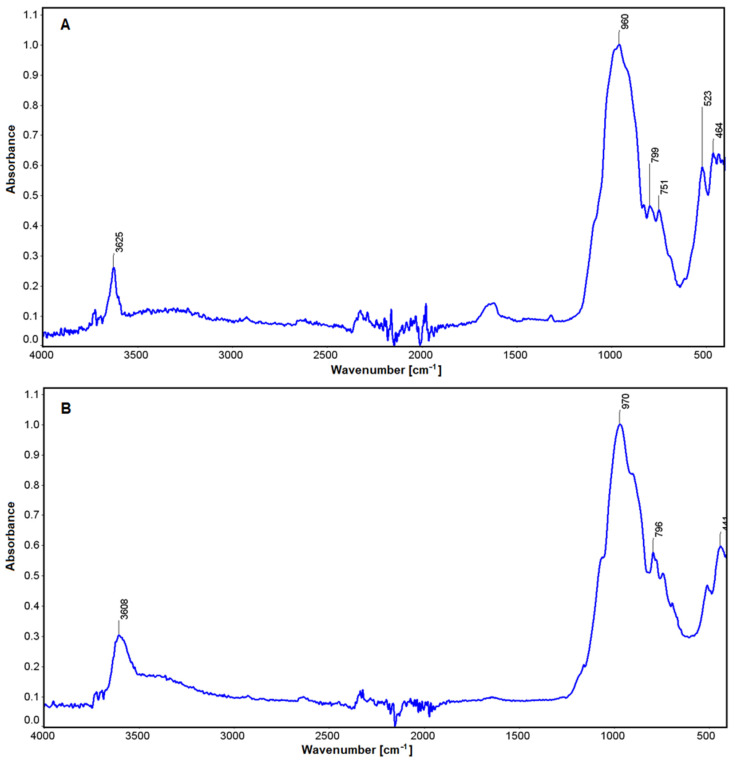
IR spectra for lepidolite (**A**) and fuchsite (**B**).

**Figure 10 materials-16-04894-f010:**
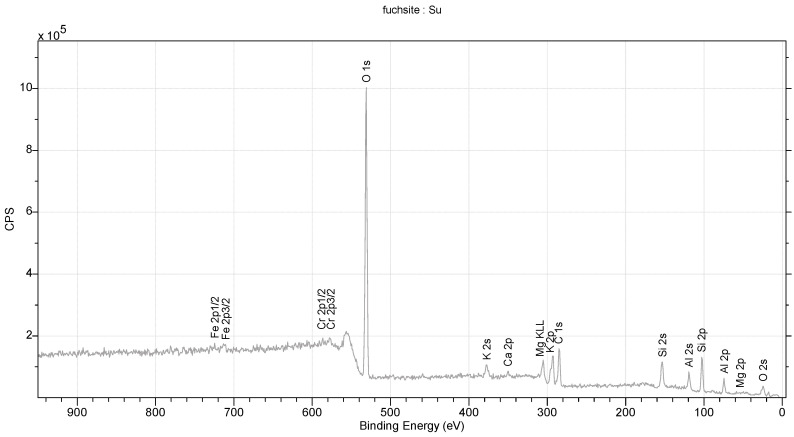
XPS spectrum for fuchsite.

**Figure 11 materials-16-04894-f011:**
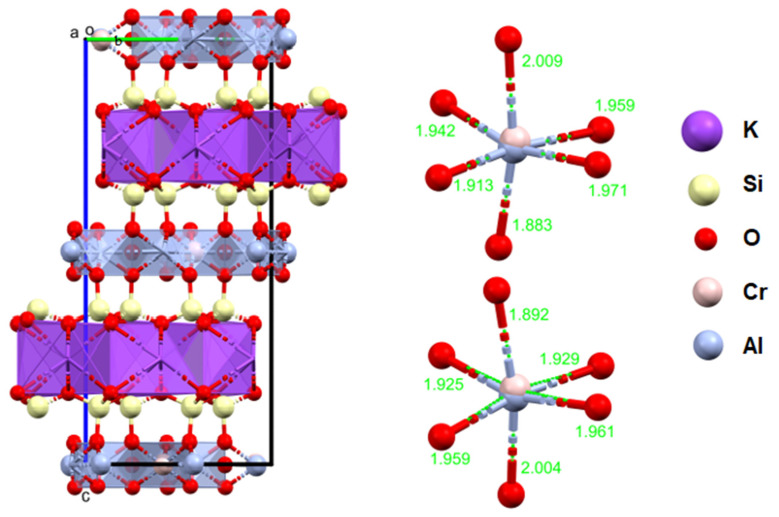
Crystal structure of the fuchsite from the analyzed rocks, as received by the authors (a, b, c—the main structural axes).

**Figure 12 materials-16-04894-f012:**
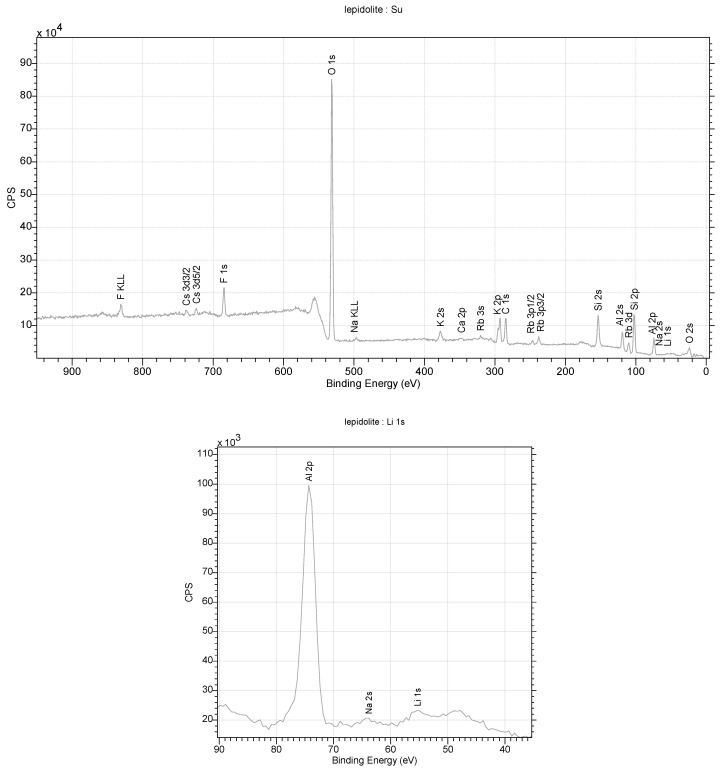
XPS spectra obtained for lepidolite from the studied rocks.

**Figure 13 materials-16-04894-f013:**
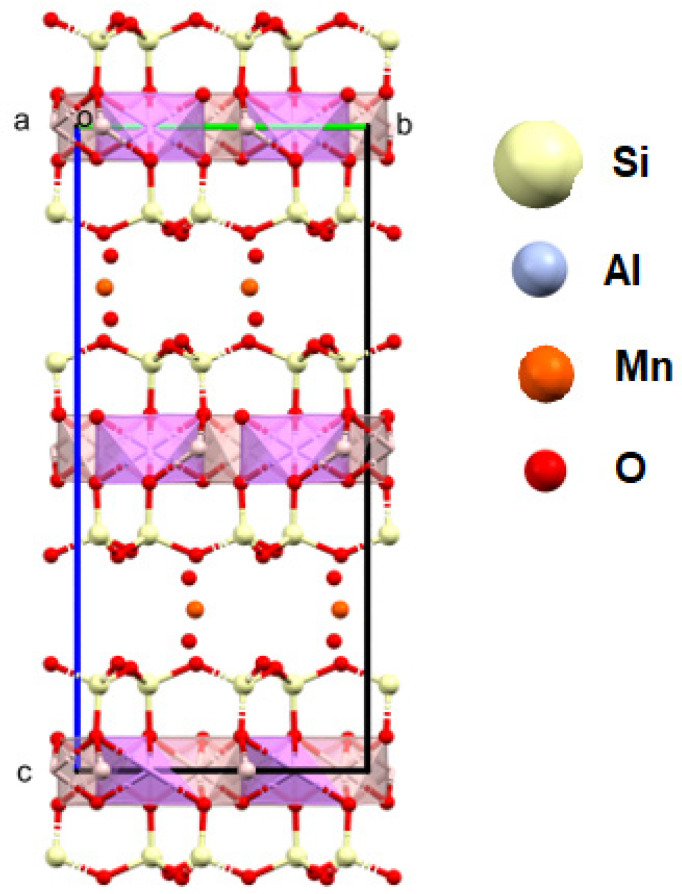
Crystal structure of lepidolite from the studied lithologies (a, b, c—the main structural axes).

**Figure 14 materials-16-04894-f014:**
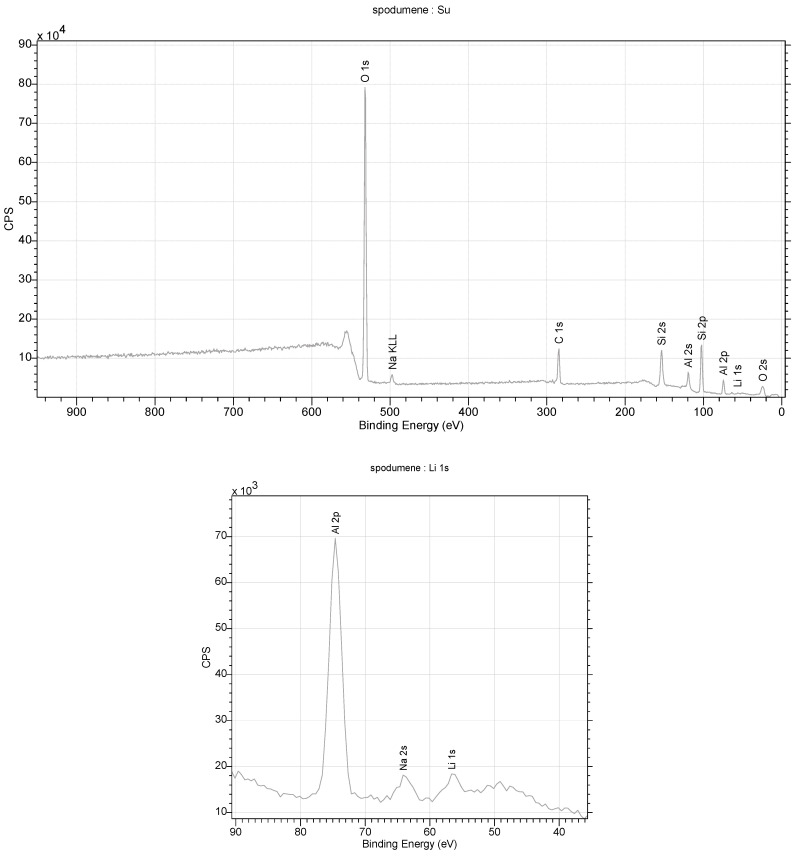
XPS spectra were obtained for spodumene from the studied rocks.

**Figure 15 materials-16-04894-f015:**
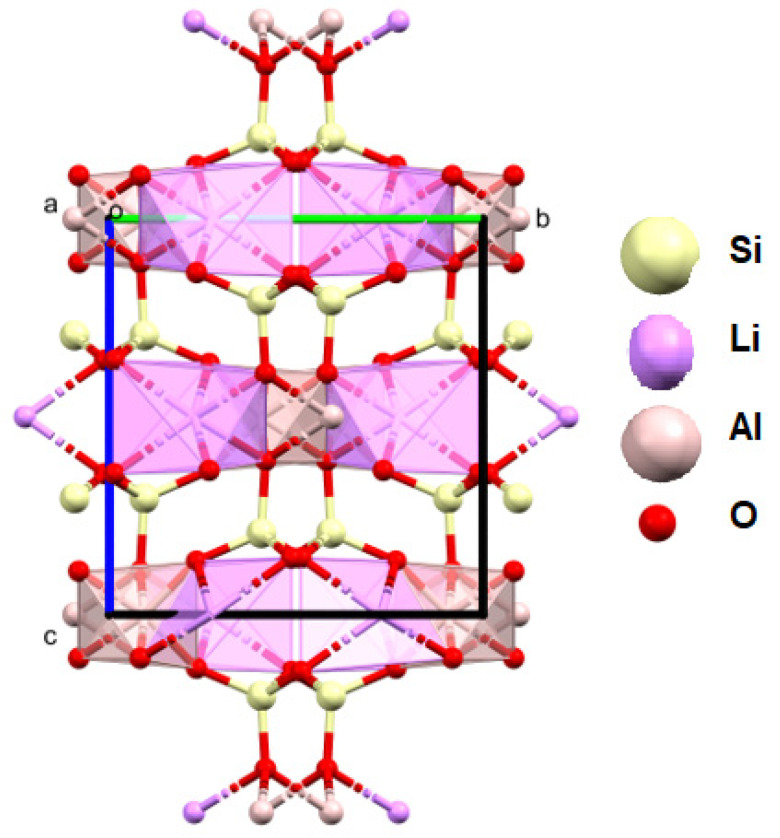
Crystal structure of spodumene (a, b, c—the main structural axes).

**Table 1 materials-16-04894-t001:** Single crystals’ X-ray diffraction results for all studied minerals.

Mineral	Fuchsite	Lepidolite	Spodumene
Empirical formula	Al_1.63_Cr_0.37_KSi_4_O_12.1_	Al_2_Fe_0.93_H_0.25_LiSi_4_O_12.12_	Al_2_Li_2_Si_4_O_12_
Temperature/K	294	294	294
Crystal system	monoclinic	monoclinic	monoclinic
Space group	C2/c	C2/c	I2/a
a/Å	5.2149(1)	5.2037(2)	5.2204(1)
b/Å	9.0505(2)	9.0250(4)	8.3947(2)
c/Å	20.0457(5)	20.1588(8)	9.0993(2)
α/°	90	90	90
β/°	95.78(1)	95.52(1)	102.40(1)
γ/°	90	90	90
Volume/Å^3^	941.29(4)	942.32(7)	389.462(17)
Z	4	4	2
ρ_calc_g/cm^3^	2.881	2.956	3.174
Crystal size/mm^3^	0.6 × 0.6 × 0.08	0.4 × 0.3 × 0.05	0.3 × 0.2 × 0.1
Data/restraints/parameters	844/0/89	854/0/98	344/0/48
Goodness-of-fit on F^2^	1.081	1.111	1.113
Final R indexes [I ≥ 2σ (I)]	R_1_ = 0.0530, wR_2_ = 0.1444	R_1_ = 0.0712, wR_2_ = 0.2039	R_1_ = 0.0184, wR_2_ = 0.0553
Largest diff. peak/hole/e Å^−3^	0.88/−0.79	0.88/−1.40	0.28/−0.26

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
