# Peer review of "The Optical and Spectroscopic Properties of Fuchsite, Spodumene, and Lepidolite from Northern Scandinavia (Kautokeino, Kaustinen, Kolmozero)"

_materials, 2023, doi:10.3390/ma16144894_

Round 1

Reviewer 1 Report

This manuscript cannot be accepted. The main problem is poor and not understandable nonprofessional terminology.

Anyway, I made some corrections for a future:

(1) Geology, microscopic description and discussion should be improved and rewrite

(2) I have several questions: geology and tectonic position of pegmatite bodies, zonation, Influence of host rocks, source, main stages of pegmatite origin: residual melt and hydrothermal solution?

(3) Main references for pegmatites of LCT family:  ÄŒerný and Ercit (2005), Linnen and Cuney (2005)

(4) Authors confuse the meaning of structure and texture. The structure - general features of the larger constituent parts of the rook developed during its formation (large scale)

The texture is determined by the size, shape, relationship and packing arrangement of different mineral grains (small scale);

In some countries, e.g Russia structure and texture have opposite meaning

Line 39 Archean

Line 40 Crystalline pedestal is not good, maybe basement

Line 43 amphibolite and granulite facies not phases

Line 48 Lithic mineralization should be lithium mineralization

Line 59 One author or all?

Line 96 Table fuchsite, not fuchsyte

Line 105 granitoid

105-106 granitoid and pegmatite intrusion should be granitic intrusion and pegmatite

Line 108-109 texture not structure

Line 110 “specimens” should be aggregates or grains

Line 112 “reticular approximation”?

Line 116 structure

Line 118 Non-transparent, if it is biotite better to use “dark minerals”, because ore minerals are also nontransparent  

Line 129 unstructured texture?

Line 131-132 Quartz has wavy (undulose) extinction

Line 131 “In the rock in question” is not good and often repeated, better “studied, investigated, exanimated, etc.

Line 132-133 “highlighted by microcline, usually in the form of vein–like zones and its corners” rewrite

Line 135 “specimens” Under the microscope, should be tabular grains, laths, aggregates

Line 137-138 “crystallizing in its vicinity” rewrite

Line 139 ‘found in gaps” -between.

Line 141 Spodumene forms compact crystals with jagged boundaries in which quartz and muscovite are visible

Line 142 “Accompanying it, hypersthene forms crystals with prominent approximations and striped structures. Non-transparent minerals are present in small numbers in the form of small crystals in the vicinity of quartz, sometimes also forming solid inclusions accompanying zircon” rewrite

Line 151 “pegmatites in question”

Line 151-152  pegmatites are associated  with fault activity in the area? Need explanation.

Line 155 dimming wavy -extinction

Line 156 “pseudo-lens character” What is mean?

Line 157-159 “Alongside the tailings are much smaller crystals of acid plagioclase, arranged in a clumped fashion, accompanying the quartz aggregates. In the rock in question, orthoclase, usually reaching considerable size, is also visible in the vicinity of the other leucocratic minerals” rewrite.

“tailings, vicinity” - should be more professional, for example- what is mean in vicinity? Under microscope relationship between mineral grains can be described. It helps to distinguish mineral formation sequence. 

Line 164 “specimens”?

Line 165” This pyroxene reaches sizes that macroscopically reach several centimeters”.?

Line 167 “opaque minerals accompanying muscovite flakes” Is it mean that ore mineral is formed together with muscovite or occurs only spatially?

Line 176 mica

Line 198 outliers?

Line 213 andesite is rock, andesine -plagioclase

Line 221 minerals  

Line 232-233 “This mineral usually has a light green color, standing out in the rock against a background of quartz and mica”?

Line 235 “diablastic approximations” What is men? Texture?

Line 236 “company of micas”?

Line 239 “darkening obliquely”- oblique extinction

Line 249-250 In the spaces between the flakes, small admixtures of rutile can be seen arranged along the schist?

Line 252 “The mineral dims straight” Parallel extinction 

Extinction could be: parallel, symmetrical, inclined, undulose  

Line 252-253 “Ripple dimming”  

Fig. 6 A- don’t need

Fig.8 what is difference between lepidolite and fuchsite?

Line 383 “residual fluid” what is mean: residual melt, residual liquid

Line 383 “mismatched”- incompatible

Line 386 “surrounding”-  host rocks

This manuscript cannot be accepted. The main problem is poor geology and not understandable nonprofessional terminology.

Anyway, I made some corrections for the future:

(1) Geology, microscopic description and discussion should be improved and rewritten.

(2) I have several questions: geology and tectonic position of pegmatite bodies, zonation, Influence of host rocks, source, main stages of pegmatite origin: residual melt and hydrothermal solution?

(3) Main references for pegmatites of LCT family:  ÄŒerný and Ercit (2005), Linnen and Cuney (2005)

(4) Authors confuse the meaning of structure and texture. The structure - general features of the larger constituent parts of the rook developed during its formation (large scale)

The texture is determined by the size, shape, relationship and packing arrangement of different mineral grains (small scale);

In some countries, e.g Russia structure and texture have opposite meaning

Line 39 Archean

Line 40 Crystalline pedestal is not good, maybe the basement

Line 43 amphibolite and granulite facies not phases

Line 48 Lithic mineralization should be lithium mineralization

Line 59 One author or all?

Line 96 Table fuchsite, not fuchsyte

Line 105 granitoid

105-106 "Granitoid and pegmatite intrusion" should be granitic intrusion and pegmatite

Line 108-109 texture not structure

Line 110 “specimens” should be aggregates or grains

Line 112 “reticular approximation”?

Line 116 structure

Line 118 Non-transparent, if it is biotite better to use “dark minerals” because ore minerals are also nontransparent  

Line 129 unstructured texture?

Line 131-132 Quartz has wavy (undulose) extinction

Line 131 “In the rock in question” is not good and often repeated, better “studied, investigated, exanimated, etc.

Line 132-133 “highlighted by microcline, usually in the form of vein-like zones and its corners” rewrite

Line 135 “specimens” Under the microscope, should be tabular grains, laths, aggregates

Line 137-138 “crystallizing in its vicinity” rewrite

Line 139 ‘found in gaps” -between.

Line 141 Spodumene forms compact crystals with jagged boundaries in which quartz and muscovite are visible

Line 142 “Accompanying it, hypersthene forms crystals with prominent approximations and striped structures. Non-transparent minerals are present in small numbers in the form of small crystals in the vicinity of quartz, sometimes also forming solid inclusions accompanying zircon” rewrite

Line 151 “pegmatites in question”

Line 151-152  Pegmatites are associated  with fault activity in the area? Need explanation.

Line 155 dimming wavy -extinction

Line 156 “pseudo-lens character” What is mean?

Line 157-159 “Alongside the tailings are much smaller crystals of acid plagioclase, arranged in a clumped fashion, accompanying the quartz aggregates. In the rock in question, orthoclase, usually reaching considerable size, is also visible in the vicinity of the other leucocratic minerals” rewrite.

“tailings, vicinity” - should be more professional, for example- what is mean in vicinity? Under the microscope relationship between minerals/grains can be described. It helps to distinguish mineral formation sequences. 

Line 164 “specimens”?

Line 165” This pyroxene reaches sizes that macroscopically reach several centimeters”.?

Line 167 “opaque minerals accompanying muscovite flakes” Is it mean that ore mineral is formed together with muscovite or occurs only spatially?

Line 176 mica

Line 198 outliers?

Line 213 andesite is rock, andesine -plagioclase

Line 221 minerals  

Line 232-233 “This mineral usually has a light green color, standing out in the rock against a background of quartz and mica”?

Line 235 “diablastic approximations” What are men? Texture?

Line 236 “Company of micas”?

Line 239 “darkening obliquely”- oblique extinction

Line 249-250 "In the spaces between the flakes, small admixtures of rutile can be seen arranged along the schist"?

Line 252 “The mineral dims straight” Parallel extinction 

Extinction could be: parallel, symmetrical, inclined, undulose  

Line 252-253 “Ripple dimming”  

Fig. 6 A- don’t need

Fig.8 What is the difference between lepidolite and fuchsite?

Line 383 “residual fluid” what is mean: residual melt, residual liquid

Line 383 “mismatched”- incompatible

Line 386 “surrounding”-  host rocks

Author Response

Dear Reviewer, thank you very much on behalf of the authors for their time and review of our text. We admit that it was written in a hurry and we missed many mistakes, for which we sincerely apologize. Our text has been rewritten and we hope you are now satisfied with the new version. We will try to upload it to the system in the next few days. Here are the point-by-point answers to your review:

(1) Geology, microscopic description and discussion should be improved and rewrite

The geology has been supplemented (extended) as well as the microscopic description (corrected) and we have also made thorough corrections in the discussion.

(2) I have several questions: geology and tectonic position of pegmatite bodies, zonation, Influence of host rocks, source, main stages of pegmatite origin: residual melt and hydrothermal solution?

This issue is basically included in the previous one, but yes - we paid attention to this question and described it in the text.

(3) Main references for pegmatites of LCT family:  ÄŒerný and Ercit (2005), Linnen and Cuney (2005)

We got to these items and got acquainted with them.

(4) Authors confuse the meaning of structure and texture. The structure - general features of the larger constituent parts of the rook developed during its formation (large scale)

The texture is determined by the size, shape, relationship and packing arrangement of different mineral grains (small scale);

The authors have corrected these descriptions throughout the text.

Below you have marked the typos that appeared in the text:

Line 39 Archean -we have changed it

Line 40 Crystalline pedestal is not good, maybe basement -we have changed it

Line 43 amphibolite and granulite facies not phases -we have changed it

Line 48 Lithic mineralization should be lithium mineralization -we have changed it

Line 59 One author or all? -we have changed it

Line 96 Table fuchsite, not fuchsyte -we have changed it

Line 105 granitoid -we have changed it

105-106 granitoid and pegmatite intrusion should be granitic intrusion and pegmatite -we have changed it

Line 108-109 texture not structure -we have changed it

Line 110 “specimens” should be aggregates or grains -we have changed it

Line 112 “reticular approximation”? -we was checked it

Line 116 structure -we have changed it

Line 118 Non-transparent, if it is biotite better to use “dark minerals”, because ore minerals are also nontransparent  We decided for the “non transparent” phrase.

Line 129 unstructured texture? -we was checked it

Line 131-132 Quartz has wavy (undulose) extinction -we have changed it

Line 131 “In the rock in question” is not good and often repeated, better “studied, investigated, exanimated, etc. -we have changed it

Line 132-133 “highlighted by microcline, usually in the form of vein–like zones and its corners” rewrite -we have changed it

Line 135 “specimens” Under the microscope, should be tabular grains, laths, aggregates -we have changed it

Line 137-138 “crystallizing in its vicinity” rewrite -we have changed it

Line 139 ‘found in gaps” -between. -we have changed it

Line 141 Spodumene forms compact crystals with jagged boundaries in which quartz and muscovite are visible. -we have changed it

Line 142 “Accompanying it, hypersthene forms crystals with prominent approximations and striped structures. Non-transparent minerals are present in small numbers in the form of small crystals in the vicinity of quartz, sometimes also forming solid inclusions accompanying zircon” we rewrote it

Line 151 “pegmatites in question” -we have changed it

Line 151-152  pegmatites are associated  with fault activity in the area? -we have changed it.

Line 155 dimming wavy -extinction -we was removed it

Line 156 “pseudo-lens character” What is mean? we rewrote it

Line 157-159 “Alongside the tailings are much smaller crystals of acid plagioclase, arranged in a clumped fashion, accompanying the quartz aggregates. In the rock in question, orthoclase, usually reaching considerable size, is also visible in the vicinity of the other leucocratic minerals” we rewrote it.

“tailings, vicinity” - should be more professional, for example- what is mean in vicinity? Under microscope relationship between mineral grains can be described. It helps to distinguish mineral formation sequence. we rewrote it

Line 164 “specimens”? -we have changed it

Line 165” This pyroxene reaches sizes that macroscopically reach several centimeters”.? we rewrote it

Line 167 “opaque minerals accompanying muscovite flakes” Is it mean that ore mineral is formed together with muscovite or occurs only spatially? we rewrote it

Line 176 mica -we have changed it

Line 198 outliers? -we have changed it

Line 213 andesite is rock, andesine -plagioclase -we have changed it

Line 221 minerals  -we have changed it

Line 232-233 “This mineral usually has a light green color, standing out in the rock against a background of quartz and mica”? we rewrote it

Line 235 “diablastic approximations” What is men? Texture? we rewrote it

Line 236 “company of micas”? we rewrote it

Line 239 “darkening obliquely”- oblique extinction -we have changed it

Line 249-250 In the spaces between the flakes, small admixtures of rutile can be seen arranged along the schist? we rewrote it

Line 252 “The mineral dims straight” Parallel extinction we rewrote it

Extinction could be: parallel, symmetrical, inclined, undulose  

Line 252-253 “Ripple dimming”   we rewrote it

Fig. 6 A- don’t need nie zgadzam siÄ™ z tym. All minerals described in the text have photographs in unpolarized and polarized light.

Fig.8 what is difference between lepidolite and fuchsite? To chyba chodziło o fig. 9. There are differences and they have been discussed in the text and information on the absorption bands has been marked in the figures

Line 383 “residual fluid” what is mean: residual melt, residual liquid melt -we have changed it

Line 383 “mismatched”- incompatible -we have changed it

Line 386 “surrounding”-  host rocks -we have changed it

Dear Reviewer, thank you very much for your help in editing our text, we apologize for the situation and we hope that the current version of the text will be more satisfactory. Authors.

Reviewer 2 Report

I revise the manuscript entitled “Optical and Spectroscopic Properties of Fuchsite, Spodumene, Lepidolite from Northern Scandinavia (Kautokeino, Kaustinen, Kolmozero)”, it is well written and understandable. I have several comments before publishing the manuscript:

1. Kautokeino location is not present in Fig. 1.

2. Defining all the abbreviations when mentioned at the first time, such as: LCT, XPS, FWHM, ……

3. The introduction section is short and without references.

4. Adding scale and altitudes to Fig. 1.

5. Some tables, figures and statements need references.

6. I suggest deleting the term “in question” wherever it is mentioned.

7. The description should be systematic along the manuscript, for example starting with Spodumene then fuchsite and then Lepidolite in all properties and methods.

8. I suggest writing the type of vibrations on the peak of the spectrum.

9. Uniform using either “Figure” or “Fig.” along the manuscript.

10. What about Li in the Spodumene at Table 2A?

11. I have additional comments in the PDF attached file.

Author Response

Dear reviewer, thank you very much for your positive opinion of our text. Thank you for taking the time to read it and assisting in the editing of our manuscript. Below we respond point by point to the proposed changes.

  1. Kautokeino location is not present in Fig. 1. -we have changed it
  2. Defining all the abbreviations when mentioned at the first time, such as: LCT, XPS, FWHM, …… We have wrote these abbreviations
  3. The introduction section is short and without references. We rewrote them
  4. Adding scale and altitudes to Fig. 1. in this drawing we have added the scale, and longitude and latitude.
  5. Some tables, figures and statements need references. All maps and diagrams have references. In other cases, where there are no references, they are the property of the authors and were created as a result of conducted analyses.
  6. I suggest deleting the term “in question” wherever it is mentioned. . -we have changed it
  7. The description should be systematic along the manuscript, for example starting with Spodumene then fuchsite and then Lepidolite in all properties and methods. The authors took note of this suggestion.
  8. I suggest writing the type of vibrations on the peak of the spectrum. -we have changed it
  9. Uniform using either “Figure” or “Fig.” along the manuscript. . -we have changed it
  10. What about Li in the Spodumene at Table 2A? In SEM-EDS, it is not possible to measure the Li content, as the authors have indicated.
  11. I have additional comments in the PDF attached file. Yes, this file has been read and suggested fixes have been changed.

Dear reviewer, thank you once again for your concern for the high quality of our publication and for your time. We appreciate your help, authors.

Reviewer 3 Report

My main criticism of this paper belongs in the Introduction section. Potential readers really need to see some references introducing Li-bearing pegmatites in general, then Li-pegmatites in Norway, Finland and Kola, their mineralogical characteristics, and so on.

Basically, in the Intro section you need to demonstrate what exactly made you undertake this work. 

Minor comments:

Line 39 - Archean, not "archaic".

Line 40 - crystalline basement, not pedestal.

Line 42 - What is the age of these intrusions. And also, please, enlighten your reader as to what intrusive rocks these are - gabbro, norite, diorite, granodiorite, etc.

Lines 42 and 43 - amphibolite and granulite facies, not phases.

Line 44 - drop "both" as you are listing 3 regions, not 2.

Line 45 - what is the age of mineralized pegmatite?

Line 59 - collected by a single author, in which case he/she should be named, or by multiple/all authors?

Line 62 - what are these "most important rock types"? Please specify.

Line 112 - please replace "acid" with Na-rich.

Line 143 - what exactly are these non-transparent minerals - sulfides, ilmenite, magnetite, other spinels?

Line 150 - what is the age of the Kolmozero-Voronya greenstone belt?

Figure 3 - there is no grey field, just yellow.

Figure 6 - where is lepidolite in microphoto A?

English needs some editing

Author Response

Dear Reviewer, thank you very much for your thorough analysis of our text and for your time. We sincerely apologize for errors in the text. The authors have thoroughly changed the content of the text, which will be attached to the system within a few days. Below we answer the review point by point.

My main criticism of this paper belongs in the Introduction section. Potential readers really need to see some references introducing Li-bearing pegmatites in general, then Li-pegmatites in Norway, Finland and Kola, their mineralogical characteristics, and so on. -we have changed it

Basically, in the Intro section you need to demonstrate what exactly made you undertake this work. 

Minor comments:

Line 39 - Archean, not "archaic". Thank You, we are rewrite

Line 40 - crystalline basement, not pedestal. Thank You, we are rewrite

Line 42 - What is the age of these intrusions. And also, please, enlighten your reader as to what intrusive rocks these are - gabbro, norite, diorite, granodiorite, etc. -we have changed it

Lines 42 and 43 - amphibolite and granulite facies, not phases. Thank You, we are rewrite

Line 44 - drop "both" as you are listing 3 regions, not 2. Thank You, we are rewrite

Line 45 - what is the age of mineralized pegmatite? -we have changed it

Line 59 - collected by a single author, in which case he/she should be named, or by multiple/all authors? Thank You, we are rewrite

Line 62 - what are these "most important rock types"? Please specify. -we have changed it

Line 112 - please replace "acid" with Na-rich. Thank You, we are rewrite

Line 143 - what exactly are these non-transparent minerals - sulfides, ilmenite, magnetite, other spinels? -we have changed it

Line 150 - what is the age of the Kolmozero-Voronya greenstone belt? -we have changed it

Figure 3 - there is no grey field, just yellow. Thank You, we are rewrite

Figure 6 - where is lepidolite in microphoto A? Thank You, we are rewrite

Thank you very much for caring about the high quality of our text and for the time devoted to it - authors.

Reviewer 4 Report

Although the subject of the paper is interesting, as it is focused on a very important contemporary aspect, that is Lithium, the quality of the paper is poor. The English language employed is very poor and requires major revisions, as in many cases the authors' arguments are not clear. Moreover, the terminology employed is also poor. I haven' t finished reviewing the entire manuscript, yet I was forced to stop at some point (page 8) as the manuscript requires major revisions in both language, methodology and discussion. 

Please see the attached file for details

The English language employed is very poor and requires major revisions, as in many cases the authors' arguments are not clear.

Author Response

Dear Reviewer, thank you very much for your thorough analysis of our text and for your time. We sincerely apologize for errors in the text. The authors have thoroughly changed the content of the text, which will be attached to the system within a few days. We have read the attached file with your suggestions and have made corrections. The finished manuscript will be uploaded to the system within a few days. Thank you very much again for your time. Authors

Round 2

Reviewer 1 Report

The geological setting part of the manuscript is improved

Some corrections:

Line 45 of Murmansk

Line 156 abbreviations 

The text should be edited. There are a few errors

Author Response

Dear Reviewer, thank you for your time and noticed errors.

The typos you indicated have been removed. Thank you for your help in editing our text.

Reviewer 4 Report

The authors have addressed the comments of the 1st review and the manuscript has been greatly improved.

The structure has been revised making the manuscript much easier to read and comprehend.

Please see the attached file for details on the review.

The manuscript requires minor grammar and spelling mistakes.

Please address to the attached file for details.

Author Response

Dear reviewer, the authors would like to thank you for your time and proposed corrections to our text. We appreciate your help. Typos have been removed in the text as indicated in the PDF file.